# Mortality in Hemodialysis Patients with COVID-19, the Effect of Paricalcitol or Calcimimetics

**DOI:** 10.3390/nu13082559

**Published:** 2021-07-26

**Authors:** María Dolores Arenas Jimenez, Emilio González-Parra, Marta Riera, Abraham Rincón Bello, Ana López-Herradón, Higini Cao, Sara Hurtado, Silvia Collado, Laura Ribera, Francesc Barbosa, Fabiola Dapena, Vicent Torregrosa, José-Jesús Broseta, Carlos Soto Montañez, Juan F. Navarro-González, Rosa Ramos, Jordi Bover, Xavier Nogués-Solan, Marta Crespo, Adriana S. Dusso, Julio Pascual

**Affiliations:** 1Department of Nephrology, Hospital del Mar, IMIM Hospital del Mar Medical Research Institute, RD16/0009/0013 (ISCIII FEDER REDinREN), 08003 Barcelona, Spain; mriera1@imim.es (M.R.); hcao@psmar.cat (H.C.); scollado@psmar.cat (S.C.); fbarbosa@psmar.cat (F.B.); mcrespo@psmar.cat (M.C.); julpascual@gmail.com (J.P.); 2Fundación Renal Iñigo Alvarez de Toledo, 28003 Madrid, Spain; 3Fundación Jimenez Díaz, 28040 Madrid, Spain; egonzalezpa@senefro.org; 4Fresenius Medical Care, Dirección Médica FMC, 28760 Madrid, Spain; abraham.rincon@fmc-ag.com (A.R.B.); ana.lopez@fmc-ag.com (A.L.-H.); sara.hurtado@fmc-ag.com (S.H.); laura.ribera@fmc-ag.com (L.R.); rosa.ramos@fmc-ag.com (R.R.); 5Department of Nephrology, Consorci Sanitari Alt Penedes Garraf, 08800 Barcelona, Spain; proyectologos1@gmail.com (F.D.); jjbroseta@clinic.cat (J.-J.B.); 6Department of Nephrology and Kidney Transplantation, Hospital Clinic, 08036 Barcelona, Spain; VTORRE@clinic.cat (V.T.); csoto@csg.cat (C.S.M.); 7Research Division and Department of Nephrology, Hospital Nuestra Señora de la Candelaria, 38010 Santa Cruz de Tenerife, Spain; jnavgon@gobiernodecanarias.org; 8Instituto de Tecnologías Biomédicas, Universidad de La Laguna, 38010 Tenerife, Spain; 9Red de Investigación Renal (REDINREN–RD16/0009/0022), Instituto de Salud Carlos III, 28029 Madrid, Spain; 10Department of Nephrology, Hospital Can Ruti, 08916 Barcelona, Spain; jbover@fundacio-puigvert.es; 11Department of Internal Medicine, Hospital del Mar, Institut Mar for Medical Research, CIBERFES, 08003 Barcelona, Spain; xnogues@psmar.cat; 12Bone and Mineral Research Unit, Instituto de Investigaciones Sanitarias del Principado de Asturias, 33011 Oviedo, Spain; adriana.dusso@gmail.com; 13Department of Internal Medicine, Division of Endocrinology, Metabolism and Lipid Research, Washington University School of Medicine, St. Louis, MO 63110, USA

**Keywords:** COVID-19, SARS-CoV-2, vitamin D, survival, calcitriol, calcifediol

## Abstract

Background. In COVID-19 patients, low serum vitamin D (VD) levels have been associated with severe acute respiratory failure and poor prognosis. In regular hemodialysis (HD) patients, there is VD deficiency and markedly reduced calcitriol levels, which may predispose them to worse outcomes of COVID-19 infection. Some hemodialysis patients receive treatment with drugs for secondary hyperparathyroidism, which have well known pleiotropic effects beyond mineral metabolism. The aim of this study was to evaluate the impact of VD status and the administration of active vitamin D medications, used to treat secondary hyperparathyroidism, on survival in a cohort of COVID-19 positive HD patients. Methods. A cross-sectional retrospective observational study was conducted from 12 March to 21 May 2020 in 288 HD patients with positive PCR for SARS-CoV2. Patients were from 52 different centers in Spain. Results. The percent of HD patients with COVID-19 was 6.1% (288 out of 4743). Mortality rate was 28.4% (81/285). Three patients were lost to follow-up. Serum 25(OH)D (calcidiol) level was 17.1 [10.6–27.5] ng/mL and was not significantly associated to mortality (OR 0.99 (0.97–1.01), *p* = 0.4). Patients receiving active vitamin D medications (16/94 (17%) vs. 65/191(34%), *p* = 0.003), including calcimimetics (4/49 (8.2%) vs. 77/236 (32.6%), *p* = 0.001), paricalcitol or calcimimetics (19/117 (16.2%) vs. 62/168 (36.9%); *p* < 0.001), and also those on both paricalcitol and calcimimetics, to treat secondary hyperparathyroidism (SHPTH) (1/26 (3.8%) vs. 80/259 (30.9%), *p* < 0.001) showed a lower mortality rate than patients receiving no treatment with either drug. Multivariate Cox regression analysis confirmed this increased survival. Conclusions. Our findings suggest that the use of paricalcitol, calcimimetics or the combination of both, seem to be associated with the improvement of survival in HD patients with COVID-19. No correlation was found between serum VD levels and prognosis or outcomes in HD patients with COVID-19. Prospective studies and clinical trials are needed to support these findings.

## 1. Introduction

The world is experiencing its third major epidemic of coronavirus (CoV) infection [1]. A new coronavirus-induced pneumonia, which appeared in late November 2019 in Wuhan, China, was named coronavirus 2019 disease (COVID-19) by the World Health Organization (WHO) on 11 February 2020 [2]. The clinical manifestations of COVID-19 patients were similar to those of SARS-CoV and MERS-CoV infections [3,4]. There is a wide spectrum of manifestations [5], ranging from asymptomatic patients [3] to acute respiratory distress syndrome (ARDS), which is the main cause of death [4]. ARDS is, in part, the result of the storm of cytokines and chemo-attractant molecules that follow viral infection [6], which is aggravated by age and pre-existing conditions, including hypertension and diabetes [7]. Despite the prevalence of variable clinical presentation and outcomes in COVID-19 infected HD patients [8,9,10], the main factors driving the severity of disease progression in these specific patients remain to be elucidated.

Recent evidence supports a direct association between low serum vitamin D (VD) levels with the severity of acute respiratory failure and COVID-19 prognosis [11], possibly mediated by VD anti-viral, anti-hypertensive and/or anti-inflammatory properties [12]. Meta-analyses of randomized clinical trials revealed protective effects of VD against acute respiratory infections, although effects were modest and heterogenous [13]. Airway diseases are associated with deregulated VD metabolism [14]; however, the data are contradictory [15], and more studies are required to evaluate such a possibility. In addition to circulating VD levels, interventions to attenuate the activation of the renin-angiotensin system (RAS), either with angiotensin converting enzyme inhibitors (ACEI) or angiotensin-2 receptor blockers (ARB), have been associated with improved COVID-19 clinical evolution and prognosis [16].

In HD patients, there is a high prevalence of hypertension, diabetes and cardiovascular disease, which concurs not only with systemic VD deficiency but also with a severely depressed 1-alpha-hydroxylation capacity to synthesize calcitriol, both renal and extrarenal. Uremic toxins are responsible for decreased 1-alpha-hydroxylase activity in cells other than kidney proximal tubular cells, suggesting an increased susceptibility for a faster progression of COVID-19 to acute lung injury. 

KDIGO [17] and Spanish guidelines [18] for chronic kidney disease (CKD) mineral and bone disorders support VD supplementation and/or calcitriol replacement therapy to treat secondary hyperparathyroidism in these patients. The aim of the present study was to evaluate the impact of VD status and the administration of VD/calcitriol and calcimimetics, used to treat secondary hyperparathyroidism, on the survival of a cohort of COVID-19 positive HD patients. 

## 2. Materials and Methods

### 2.1. Patients and Variables

This is a cross-sectional retrospective observational study of 288 HD patients with SARS-CoV2 positive PCR, from 52 Spanish HD facilities from 12 March 2020 to 21 May 2020. This sample is representative of the Spanish HD population, in terms of age and sex. The study was performed under the principles of the Declaration of Helsinki and was approved by the hospital ethics committee at participant institutions.

Demographics, clinical, laboratory and radiological data were obtained from medical records launched at the beginning of the epidemic. After the diagnosis, all patients were followed for at least 14 days, for a maximum of 35 days or until death. Three patients were lost to follow-up. All related-to-mortality statistical analyses were conducted in 285 patients. Parameters that were recorded included serum concentrations of 25-hydroxyvitamin-D (25-OH-D3), PTH, calcium, phosphate and treatment with 25-OH-D3 (calcifediol), active VD (calcitriol; paricalcitol) and/or calcimimetics shortly (20 days) before COVID-19 infection.

Other variables that were recorded were age; sex; race; body mass index (BMI); type of vascular access; comorbidities; baseline treatments, including angiotensin converting enzyme inhibitors (ACEIs), angiotensin receptors blockers (ARBs) or nonsteroidal anti-inflammatory drugs (NSAIDs); clinical symptoms; chest X-Ray findings; standard laboratory parameters; specific treatments for COVID-19; and outcomes. Laboratory parameters were recorded at two points: at baseline (regular control tests performed immediately prior to the time of COVID-19 diagnosis (median 20 days, interquartile range–IQR-12.5–25 days)) and at hospital admission.

### 2.2. SARS-CoV-2 Testing

Testing for SARS-CoV-2 virus infection was undertaken using qualitative reverse transcription PCR (RT-PCR), following M2000^®^ RNA Extraction Kit according to the manufacturer’s instructions (Abbott, Chicago, IL, USA). Several molecular diagnostic RT-PCR platforms were used, including Roche Cobas^®^ and Abbott^®^ Real-time SARS-CoV2 Assay^®^, (Chicago, IL, USA), which provide the quantification of viremia ≥20 copies/mL. Screening tests looked for gen *E* (envelope protein), and confirmatory genes of SARS-CoV-2 were *RdRp* (RNA-dependent RNA polymerase) and *N* (nucleocapsid protein).

### 2.3. Statistical Analysis

Data are presented as mean and standard deviation (SD) for normally distributed variables; median and IQR for non-normally distributed variables; and percentages (%) for categorical variables. 

Univariant logistic regression analyses were used to evaluate the association between circulating VD levels and hospital admission or medical care requirements.

For survival analysis, we used Log-Rank Tests to compare Kaplan–Meier probability of death curves and performed univariate and multivariable Cox regression analyses. Based on the sample size and the number of events, variables significantly related with mortality in univariate analyses were prioritized and included in a multivariate Cox model to obtain the corresponding hazard ratios (HRs) and 95% confidence intervals (CIs). The “log minus log” (LML) plots were used to assess the proportional hazards assumption for categorical variables. For quantitative variables, we created interactions of predictors and a function of survival time to generate the time-dependent covariates. Variables and their time-dependent covariates were included in Cox models to assess the proportional hazards assumption. 

Continuous predictors were categorized into quartiles and tertiles and included in multiple univariate models to assess whether linearity assumption was reasonable. Only variables with linear effects were included as continuous predictors. For predictors with non-linear effects, several cut-off values were chosen to generate balanced groups in a clinically relevant manner: age < 70, 70–80 and >80 years; vitamin D < 20, 20–30 and >30 ng/mL; C-reactive protein < 5.6, 5.6–24, >24 mg/L; leukocytes < 4500, 4500–7000 and >7000 n/mm^3^; lymphocytes < 700, 700–1200 and > 1200 n/mm^3^; ferritin < 400, 400–800 and > 800 µg/L; D-dimer <700, 700–1500 and >1500 ng/mL; LDH <200, 200–300 and >300 IU/L; and finally, procalcitonin < 0.5, 0.5–1 and >1 ng/mL. The Endocrine Society Clinical Practice Guideline on the Evaluation, Treatment and Prevention of Vitamin D Deficiency defined VD deficiency as a serum 25(OH)D level < 20 ng/mL and insufficiency between 20 and 29 ng/mL: a target of >30 ng/mL is suggested for optimal health [19,20]. 

All *p*-values were two-sided. Statistical significance was set at *p* < 0.05. Statistical analyses were performed using IBM^®^ SPSS^®^ Statistics version 19 (IBM Corp., Armonk, NY, USA). 

## 3. Results

COVID-19 was diagnosed through a positive RT-PCR test for SARS-CoV-2 in 288 out of 4743 HD patients (6.1%). Demographic parameters, comorbidities and baseline treatments and analytics in COVID-19 HD patients are shown in Table 1. Median [p25–75] [percentile 25-75]serum 25-OH-VD (calcidiol) level was 17.1 [10.6–27.5], with 43.6% of patients with levels <20 ng/mL, 36.5% of patients within the 20–30 ng/mL range and only 19.9% with levels higher than >30 ng/mL. The outcomes, clinical profile and treatments of COVID-19 maintenance hemodialysis patients are presented in Table 2. By the end of the follow-up period, 28.4% (81/285) of patients had died. The most frequent symptoms were fever (67.9%), cough (52.4%) and dyspnea (32%). Only 12.6% of our patients were asymptomatic. Seventy-three per cent of patients with chest X-ray in our cohort presented with pneumonia (194/265), and 80% underwent hospital admission. Regarding specific treatments for COVID-19 patients (Table 2), hydroxychloroquine (73.9%), azithromycin (39.3%), lopinavir/ritonavir (8.7%), steroids (32.3%), tocilizumab (8.7%) and beta-interferon (5%), were the most frequently prescribed drugs. Ten percent of patients did not receive any specific treatment. 

### Serum VD Levels and Outcomes

No significant associations were observed between circulating 25-OH-VD levels and hospital or intensive care unit (ICU) admission, non-invasive/mechanical ventilation requirements (Appendix A) or mortality (Log Rank (Mantel-Cox) 0.341; *p* = 0.843).

Variables that were associated to mortality also included treatment of secondary hyperparathyroidism.

In order to elucidate the possible risk factors which could lead to a fatal progression in hemodialysis patients, we performed different univariate Cox regression analyses. An increase in age or in the serum levels of inflammatory markers (PCR, CK, D-dimer and procalcitonin), as well as the appearance of dyspnea or pneumonia, was significantly associated with a fatal outcome. However, other factors, such as PTH and albumin levels, BMI, and, interestingly, previous treatment with paricalcitol or calcimimetics, were associated with increase survival. (Table 3) (Figure 1). Within the group of patients treated with calcitriol, mortality was not different in patients with VD deficiency or insufficiency, compared to patients with VD sufficiency (Data not shown). 

Mortality associated with COVID-19 was significantly lower in patients treated with any form of vitamin D (27/136, 19.9%), compared to those that were untreated (54/149; 36.2%) *p*: 0.002. Only patients treated with paricalcitol or calcimimetics showed significantly lower mortality, compared to those who received calcidiol or calcitriol (Table 3) (Appendix A). 

Then, we compared possible differences between baseline characteristics of COVID-19 HD patients treated with no active vitamin D medication, paricalcitol, calcimimetics, paricalcitol or calcimimetics, to those who were not treated with either drug (Appendix A, respectively). Patients treated with paricalcitol and/or calcimimetics had a significantly higher serum PTH; while phosphate was significantly higher in patients treated with calcimimetics. Calcimimetics treated patients were younger and their dialysis vintage was lower than those who did not receive them. No significant differences between treated and non-treated patients were observed in hospital admission, non-invasive mechanical ventilation or endotracheal intubation. (Appendix A). 

The multivariate Cox regression model displays some independent predictive factors for fatal outcomes: age 70–80 y.o. (HR 2.911; 95%CI 1.532–5.532), age > 80 y.o. (HR 2.372; 95%CI 1.195–4.707) and PCR > 24 mg/L (HR 2.417; 95%CI 1.284–4.55) were independent risk factors for mortality in COVID-19 HD patients. Cox regression analysis confirmed an increased survival in HD patients treated with paricalcitol (HR 0.499; 95%CI 0.278–0.898) and/or calcimimetics (HR 0.332; 95%CI 0.12–0.921) and with higher serum albumin levels (HR 0.668; 95%CI 0.496–0.9) (Table 4).

## 4. Discussion

Among the classical risk factors of COVID-19 associated mortality in the general population [3], only age was independently associated with mortality in our cohort of Spanish HD patients. Common biomarkers of severity and/or poor prognosis also markedly differed. Moreover, we did not find a significant negative association between serum baseline 25-OH-VD levels prior to the diagnosis of COVID-19 with mortality in our HD cohort. However, we did find a remarkable association with improved survival, not only with an active VD analog (paricalcitol) treatment, but also with the use of calcimimetics. Therefore, we describe that the impact of COVID-19 mortality risk prognostic markers is different in this specific population, and that the alleged and well described plausible benefits of VD pleiotropic effects for COVID-19 may be extended to other anti-hyperparathyroidism agents, such as calcimimetics. 

The prevalence of confirmed COVID-19 in our cohort was 6.1%, lower than that described in other studies (range between 10–40%) [7,8,9]. There is a heterogeneous prevalence and mortality rate, according to different geographic areas, including within Spain [21]. The average number of patients undertaking HD with diabetes mellitus (DM) is around a 16–17% in the Spanish Registry. In the Spanish COVID-19 registry, 22% of COVID-19 patients had DM. The prevalence of DM in this cohort is higher than that reported for the whole country. It is important to emphasize that there are major differences in DM according to the geographic area of the country, and this particular HD cohort includes the Canary Islands, one of the areas with the highest incidence of DM in Spain [21]. Similar to other series, the most frequent symptoms observed were fever, cough and dyspnea [7,8,9,22]. Laboratory findings, incidence of pneumonia, hospital admission and mortality rates in our cohort are consistent with other published data on COVID-19 HD patients [7,8,9]. 

Phosphate (4.1 mg/dL), calcium (8.8 mg/dL) and PTH (240 pg/dL) levels within limit reflects an optimal control of MBD in this cohort. [17,18]. Patients treated with calcimimetics were younger and their dialysis vintage was lower than those who did not receive them. Dialysis vintage is a key determinant of the severity of secondary hyperparathyroidism. A direct relation between dialysis vintage and the severity of hyperparathyroidism has been demonstrated [23]. On the other hand, difficulties for the treatment of SHPT in clinical practice are frequently encountered, from the occasional inadequacy on dosage by the clinicians or the frequent non-compliance to the therapy by CKD patients. This is more frequent in younger patients, in which the use of calcimimetics could the best option, as hyperphostatemia is a limitation of the use of vitamin D analogs [24]. In fact, serum phosphate levels were higher in patients treated with calcimimetics.

VD deficiency is highly prevalent in patients on dialysis, and a low serum 25(OH)D level has been significantly associated with a higher risk of all-cause and cardiovascular mortality [25]. Some authors defend the hypothesis that low VD status is a consequence of poor health, rather than its cause [26]. Nevertheless, recent reviews hypothesized that VD insufficiency may compromise respiratory immune function, thereby increasing the risk of COVID-19 severity and mortality [27]. Some retrospective studies demonstrated a correlation between VD status and COVID-19 severity and mortality [28], while other studies did not find any correlation after the adjustment for confounding variables [29,30]. In our study, we did not find significant associations between circulating 25-OH-VD levels and outcomes in COVID-19 HD patients, including hospital admission, ICU admission, requirement of noninvasive mechanical ventilation, time from admission to resolution, and mortality, similar to that shown in other studies [28]. One potential explanation is the well-recognized population-dependent requirements of circulating 25(OH)D levels for the efficacious health benefits of non-classical vitamin D actions, including defeating COVID-19 infection and decreasing the risk of hospitalization. Indeed, while in the general population, vitamin D deficiency increases by 45% the risk of infection and by 95% the risk of hospitalization [31], circulating levels of calcidiol from 30 to 40 ng/mL cannot protect otherwise healthy African Americans from COVID-19 infection. Their risk of infection with these serum concentrations remains 2.45-fold higher than individuals with serum calcidiol >40 ng/mL [32]. Therefore, it is critical to personalize the levels of calcidiol in hemodialysis patients, to confer effective protection against COVID-19. Medical record keeping during the pandemic should help provide a first step in achieving an efficacious correction of vitamin D deficiency in Covid19 patients.

An effective attenuation of COVID-19 disease severity from adequate VD supplementation, or from treatment with VD receptor activators (calcitriol, paricalcitol) may be expected in HD patients. The arguments for this are the well-recognized VD anti-inflammatory, anti-viral, anti-hypertensive properties and its seemingly efficacious protection against respiratory infections [13], in addition to active VD efficaciously attenuating the progression of experimental renal or cardiovascular disease [30,33,34,35,36]. Moreover, a recent pilot study performed on non-CKD patients suggested that the administration of a high dose of calcifediol (25-OH-D3) reduced the need for intensive care treatments of COVID-19 patients requiring hospitalization [37].

An important finding in our study is the significantly lower COVID-19 mortality in patients treated with VD. Remarkably, only patients treated with paricalcitol but not calcitriol or calcifediol, showed a significantly lower mortality; however, the number of patients treated with calcitriol or calcifediol was small [11], and, therefore, we may lack sufficient power to discriminate between these two forms of active and moderately active endogenous VD metabolites.

From a mechanistic point of view, there are good reasons to postulate that VD may favorably modulate host responses to SARS-CoV-2, both in the early viraemic and later hyper-inflammatory phases of COVID-19. Regarding potential VD specific actions against COVID-19, calcitriol inhibits Skp2, the protein induced by SARS-COV2 to hijack Beclin1-driven autophagy, inhibits viral replication [38]. An additional and critical calcitriol action against COVID-19 is the induction of ACE2, which the downregulation by SARS-CoV-2 exacerbates the adverse impact on the lungs of the excessive inflammatory response, initiated by the high activity of the renin–angiotensin II system [39,40,41]. ACE2 converts angiotensin II into angiotensin 1–7, which binds its Mas receptor to initiate anti-fibrotic, anti-inflammatory signals that protect lung integrity by counteracting the deleterious effects of RAS activation [42]. The inhibition of ADAM17 expression and activity by VD receptor activators may contribute to attenuating ACE2 shedding, an ADAM17-driven cleavage; in addition, it also reduces the pro-inflammatory and pro-fibrotic signals downstream from angiotensin II-induced increases in ADAM17 activity at the cell surface of viral infected cells [43]. Pre-existing hypertension increases the risk of ICU admission, mechanical ventilation and death in patients with COVID-19. Calcitriol or paricalcitol simultaneous suppression of renin and induction of ACE2 may support a protective effect on lung injury, attenuating mortality. Paricalcitol pro-survival benefits in COVID-19 HD patients were dose independent and did not differ between VD deficient or sufficient individuals. In a nested case-control analysis in dialysis patients, low VD levels were associated with increased mortality, but a significant interaction was noted between VD levels, subsequent active VD therapy (mainly paricalcitol) and survival [42]. Patients treated during their first 9–12 months of hemodialysis with active forms of VD (mainly paricalcitol) had better survival, compared to those untreated, regardless of 25-OH-VD levels [44]. A potential survival advantage of paricalcitol over calcitriol in HD patients is also possible [45], but the mechanisms responsible for the different efficacy of these two VD receptor activators remain incompletely characterized.

Active VD targeting of ADAM17 [46] and consequently, the attenuation of ACE2-cleavage to maintain its protective pathways, may not be able to fully answer the questions regarding COVID-19 protection [47]. Unexpectedly, our results from univariate and multivariate models also suggest a protective effect of calcimimetic treatment on mortality in HD patients. The anti-hypertensive [48,49] and FGF-23 [46] suppressive properties of calcimimetic drugs may contribute to their survival advantage in COVID-19 patients. The benefits of attenuating RAS activation have been already discussed for VD and paricalcitol. On the other hand, extremely high levels of FGF-23 have been described in HD patients, and have been strongly associated with mortality [50]. Therefore, the proven efficacy of calcimimetics in reducing high FGF-23 levels could have played a role in attenuating either the cardiac damage [51], or FGF23-mediated suppression of macrophage calcitriol production [52] necessary for autocrine-paracrine anti-viral and anti-inflammatory actions by immune cells. It is possible that calcimimetics also mitigate the increases in macrophage TNFα, thereby attenuating the cytokine storm associated with COVID-19 [53], or preventing an impaired neutrophil activation to mount an appropriate anti-viral response [54]. Even though all of the above are well recognized deleterious actions of high FGF-23 levels, and treatment with calcimimetics efficaciously reduced serum FGF-23 in several studies, the absence of FGF-23 measurements in this study renders these proposed mechanisms as purely speculative. On the other hand, it is well known that VD increases FGF-23, but, at the same time, VD treatment attenuates cardiac FGF3/FGR4 signaling and hypertrophy in uremic rats [33].

The cross-sectional and retrospective nature of this analysis has limitations, such as the limited patient number, the lack of all analytical variables in some patients and the inherent associated selection bias and residual confounding, thus no conclusions on causality can be drawn. The absence of randomization led to some imbalances between groups; however, proper multivariant analysis, including the best-known variables to date, should have corrected these imbalances. Neither serum 25-OH-VD nor FGF-23 levels were obtained at the time of hospital admission, but our “baseline” values may offer a different perspective of the significance of VD deficiency, shortly before COVID-19 versus during active treatment when COVID-19 is ongoing. Moreover, we underline that other commonly used drugs in HD patients may also be of value. Among the strengths of our study, we describe novel associations of COVID-19 mortality in a significant cohort of HD patients. We did not only underline that clinical and biochemical classical mortality risk factors may differ from those described for the general population, but also that other antiparathyroid agents with pleiotropic effects beyond active forms of VD may offer some degree of protection. These findings may open new perspectives and novel pathways, which may improve the care of this complex and specific population.

In summary, no association was found between serum 25-OH-VD levels, prognosis and evolution of COVID-19 in our cohort of HD patients. Paricalcitol and/or calcimimetics was associated with improved survival in the complex COVID-19 HD patients. In fact, calcimimetics and paricalcitol have well known pleiotropic effects beyond mineral metabolism, that may be of value in the current scenario of extreme mortality in this population. This remains to be proven, and prospective studies and/or clinical trials are needed to validate and extend our observations.

## Figures and Tables

**Figure 1 nutrients-13-02559-f001:**
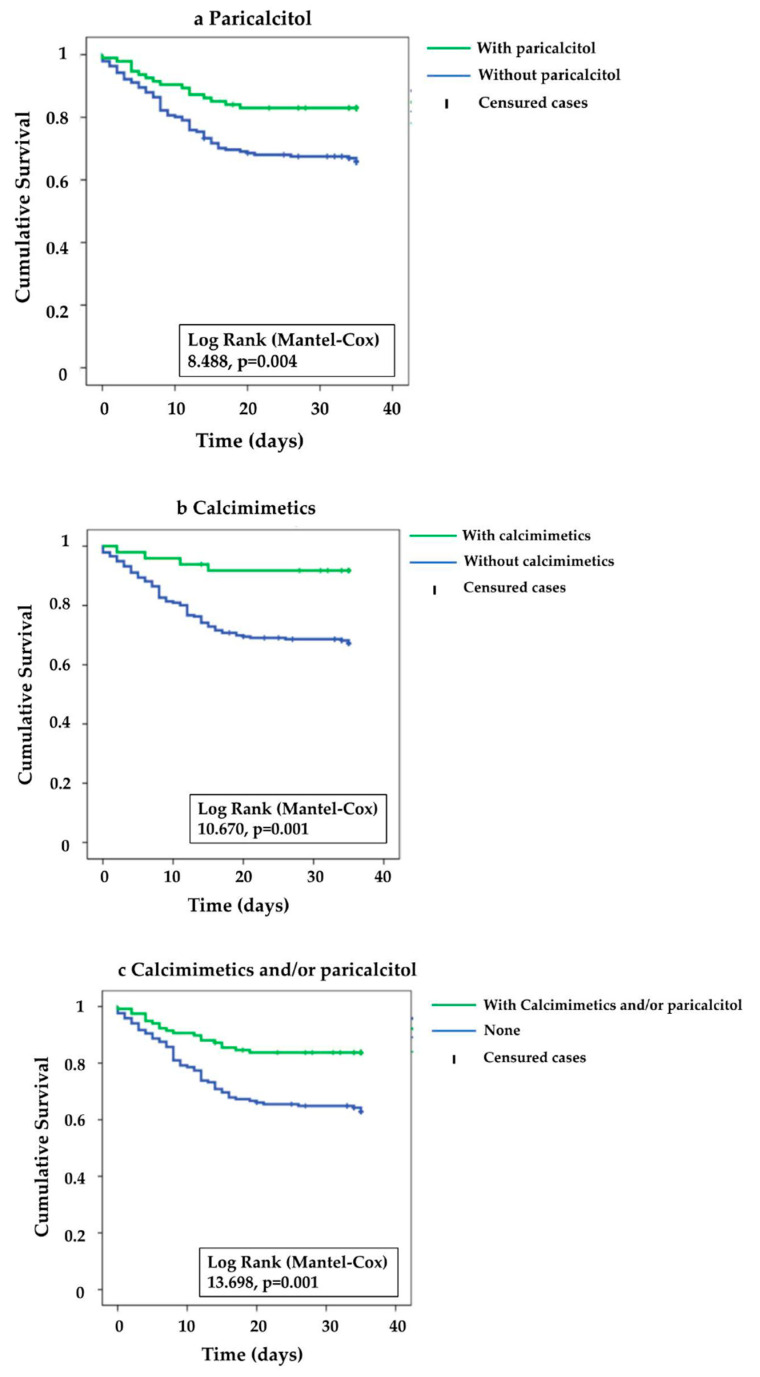
COVID-19 survival in HD patients according to their previous use of anti-secondary hyperparathyroidism agents. (**a**). With paricalcitol N:94, without paricalcitol N:194; (**b**). With calcimimetics N: 49, without calcimimetics N: 239; (**c**). With calcimimetics and/or paricalcitol N: 117, without calcimimetics and/or Paricalcitol N: 171; (**d**). With calcimimetics and/or paricalcitol N: 171, calcimimetics N: 49 and paricalcitol N:94.

**Table 1 nutrients-13-02559-t001:** Baseline characteristics of COVID-19 maintenance hemodialysis patients (N: 288).

Demography		
	Age (years, mean ± SD)	72.4 ± 12.6
	Body mass index (kg/m^2^, mean ± SD)	25.6 ± 5.5
	Time on Hemodialysis (years, median [IQR])	2.5 [1.2–5.2]
Sex		
	Female	84 (29.2)
Race		
	Caucasian	208 (72.2)
	Other	11 (3.8)
	Unknown	69 (23.9)
Vascular access (N: 261) **		
	Arteriovenous fistulae	154 (59)
	Central venous catheter	107 (41)
Cause of renal failure		
	Diabetic Nephropathy	85 (29.5)
	Vascular	37 (12.8)
	Glomerular	34 (11.8)
	Hereditary	18 (6.2)
	Tubulointerstitial	15 (5.2)
	Unknown	78 (27.1)
	Other	21 (7.2)
Toxic habits (N: 117) **		
	Current smoking	102 (49)
	Alcohol abuse	17 (9.6)
Risk factors and comorbidities (N: 288)		
	Diabetes mellitus	137 (47.6)
	Hypertension	129 (44.8)
	Dyslipemia	106 (36.8)
	Ischemic heart disease	94 (32.6)
	Lung disease	66 (22.9)
	Previous trasplantation	20 (6.9)
	Tuberculosis	5 (1.7)
Previous treatment for secondary hyperparathyroidism (N: 288)		
No treatment		128 (44.4)
Any treatment (Vitamin D and/or calcimetics)		160 (55.5)
Any calcimimetic		49 (17.0)
	Cinacalcet	33 (11.5)
	Etelcalcetide	16 (5.6)
No vitamin D treatments		151 (52.4)
Any vitamin D treatments		137 (47.6)
	Calcifediol (25-OH-Vitamin D)	33 (11.5)
	Calcitriol	22 (7.6)
	Paricalcitol	94 (32.6)
	Two kind of Vitamin D	12(4.2)
	One kind of vitamin D	125 (43.4)
Calcimimetics or paricalcitol		117 (40.6)
Both calcimimetics and paricalcitol		26 (9.0)
	Calcidiol dose (mcg/month, median, [IQR])	399 [399–798]
	Calcitriol dose (mcg/week median, [IQR])	1.75 [0.99–1.83]
	Paricalcitol dose (mcg/week median, [IQR])	4.10 [2.25–5.60]
Other concomitant medication		
	ACEIs	63 (21.9)
	ARBs	32 (11.1)
	NSAIDs	21 (7.3)
Baseline laboratory data ***		
Time from baseline Blood test and PCR positive (days, median [p25–75])		20.0 [12.5–25]
Hemoglobin (gr/dL, mean (SD))		11.0 (1.3)
White blood cells (×10 ^3^/uL, median [p25–75])		6.4 (5–8.3)
Lymphocytes (×10 ^3^/uL, median [p25–75])		1.07 (0.76–1.4)
Neutrophils (×10 ^3^/uL, median [p25–75])		4.26 (3.2–5.5)
Platelets (×10 ^3^/uL, median [p25–75])		190 (146–227)
Fibrinogen (UI/L, mean (SD))		433 (139)
AST (UI/L. median [p25–75])		16 (12–20)
Albumin (g/dL. mean (SD))		3.8 (0.3)
C-reactive protein (mg/L, median [p25–75])		7.9 (3.1–19.4)
Ferritin (mcg/mL. median [p25–75])		436 (265–586)
Transferrin Saturation Index (%. median [p25–75])		26 (17.8–36)
Calcium (mg/dL. mean (SD))		8.8 (0.7)
Phosphate (mg/dL, mean (SD))		4.1 (1.1)
PTH (pg/dL, mean (SD))		240 (133–398)
Hemoglobin (gr/dL, mean (SD))		10.9 (1.6)
White blood cells (×10 ^3^/uL, median [p25–75])		5.5 (4.2–7.7)
Lymphocytes (×10 ^3^/uL, median [p25–75])		0.70 (0.4–1.0)
Neutrophils (×10 ^3^/uL, median [p25–75])		4.0 (2.8–5.6)
Platelets (×10 ^3^/uL, median [p25–75])		166 (120–204)
D-dimer (mcg/L, median [p25–75])		982 (580–2000)
Fibrinogen (UI/L, mean (SD))		593 (194)
AST (UI/L, median [p25–75])		21 (16–28)
LDH (UI/L, mean (SD))		353 (664)
IL6 (ng/L, median [p25–75])		18 (8.9–59.9)
Albumin (g/dL, mean (SD))		3.3 (0.5)
C-reactive protein (mg/L, median [p25–75])		14.7 (4.3–61)
Ferritin (mcg/mL, median [p25–75])		1000 (547–1566)
Transferrin Saturation Index (%, median [p25–75])		20 (13.8–31)
Calcium (mg/dL, mean (SD))		8.6 (0.7)
Phosphate (mg/dL, mean (SD)		4.6 (1.5)

Data are expressed as n (%) or otherwise specified. SD: standard deviation; IQR: interquartile range; [p25–75]: percentile 25 and 75. ACEIs: angiotensin converting enzyme inhibitors; ARBs; anti-AT1 receptor antagonists, NSAIDs: non-steroidal anti-inflammatory agents. ** Variables without 100% of data. *** Baseline laboratory data: regular control tests performed immediately prior to the time of COVID-19 diagnosis (median 20 days, interquartile range–IQR-12.5–25 days).

**Table 2 nutrients-13-02559-t002:** Outcomes and clinical profiles of COVID-19 maintenance hemodialysis patients (N: 285).

**Hospital admission N, %**	
Admission required (N: 285)	228 (80)
ICU admission (N: 257)	15 (5.8)
Mechanical ventilation (N: 254)	10 (3.9)
**Outcome**	
Resolution and/or discharged (N: 285)	169 (59.2)
Exitus (N: 285)	81 (28.4)
Admission to discharge (time elapsed in days, median [p25–75])	16.0 [10.8–28.0]
Admission to exitus (time elapsed in days days, median [p25–75])	8.0 [ 4.7–14.2]
**Approached COVID-19^+^ contact**	
Nursing home or relatives (N: 285)	52 (18.2)
**Symptoms**	
Asymptomatic (N: 276)	35 (12.6)
Fever (N: 265)	180 (67.9)
Cough (N: 265)	139 (52.4)
Dyspnea (N: 265)	85(32.07)
Other (sore throat, discomfort, muscle pain, fatigue) (N: 265)	66 (24.9)
Diarrhea (N: 265)	28 (10.5)
Ageusia (N: 93)	4 (4.2)
Anosmia (N: 93)	5 (5.2)
**Lung disease** (N: 265)	
Pneumonia	194 (73.2)
**Covid-19 treatment**	
Hydroxychloroquine (N: 238)	176 (73.9)
Azithromycin (N: 239)	94 (39.3)
Lopinavir/Ritonavir (N: 239)	21(8.7)
Steroids (N:284)	92 (32.3)
Tocilizumab (N: 239)	21 (8.7)
Beta-Interferon (N: 239)	12 (5)
Remdesevir (N: 239)	1 (0.4)
Other (Antibiotics.anticoagulants..etc) (N: 239)	137 (57.3)
No treatment (N: 239)	24 (10)

Data are expressed as *n* (%) or as otherwise specified.

**Table 3 nutrients-13-02559-t003:** Clinical characteristics and associations with mortality in univariate Cox regression analyses.

	HR [CI 95%]	*p*
Demographics and Comorbidities		
Age (years) < 70 (Ref.)		
70–80	2.889 [1.569–5.321]	0.001
>80	2.817 [1.529–5.187]	0.001
Gender (female)	1.195 [0.749–1.905]	0.455
Race (caucasian)	0.957 [0.738–1.241]	0.742
Body mass index (kg/m^2^)	0.929 [0.882–0.979]	0.006
Dialysis vintage (years)	1.011 [0.974–1.050]	0.563
Renal etiology	0.983 [0.908–1.065]	0.682
Current Alcohol status	0.347 [0.084–1.429]	0.143
Current Smoking status	1.019 [0.617–1.683]	0.941
Native fistulae as vascular access	0.679 [0.428–1.078]	0.101
Previous transplant	0.915 [0.369–2.271]	0.848
Diabetes Mellitus	1.080 [0.698–1.670]	0.730
Arterial hypertension	0.974 [0.629–1.510]	0.908
Tuberculosis	0.649 [0.090–4.672]	0.668
Dyslipemia	0.839 [0.529–1.332]	0.456
Lung disease	1.537 [0.959–2.463]	0.074
Ischemic cardiopahty	1.181 [0.750–1.859]	0.474
Mineral Metabolism		
25-OH-Vitamin D. (ng/mL)	0.994 [0.975–1.012]	0.491
25-OH Vitamin D (ng/mL)		
<20 (Ref.)	1	
20–30	1.185 [0.663–2.115]	0.567
>30	1.077 [0.586–1.979]	0.811
PTH (ng/L)	0.998 [0.997–0.999]	0.002
Calcium (mg/dL)	0.875 [0.765–1.000]	0.051
Phosphate (mg/dL)	0.879 [0.758–1.020]	0.089
Vitamin D derivatives use	0.493 [0.310–0.782]	0.003
Calcidiol use	1.140 [0.591–2.222]	0.686
Calcidiol dose (mcg/month)	0.999 [0.999–1.000]	0.653
Calcitriol use	0.587 [0.215–1.605]	0.260
Calcitriol dose (mcg/week)	0.489 [0.110–2.175]	0.348
Paricalcitol use	0.455 [0.263 -0.787]	0.002
Paricalcitol dose (mcg/week)	0.915 [0.749–1.118]	0.385
Calcimimetics use	0.220 [0.080–0.601]	0.003
Other Treatments		
ACEIs	0.903 [0.523–1.560]	0.715
ARBs	1.050 [0.525–2.099]	0.891
NSAIDs	0.821 [0.332–2.029]	0.669
COVID-19 Clinical symptoms		
Ageusia	2.112 [0.499–8.945]	0.310
Anosmia	1.565 [0.370–6.625]	0.543
Cough	0.813 [0.515–1.283]	0.374
Fever	1.665 [0.998–2.779]	0.051
Diarrhea	0.832 [0.382–1.810]	0.642
Dyspnea	1.755 [1.110–2.774]	0.016
Asthenia	1.867 [1.165–2.993]	0.009
Asymptomatic	0.259 [0.082–0.821]	0.022
Pneumonia	4.705 [2.037–10.867]	<0.001
X-Ray compatible with COVID-19	5.629 [1.735–18.265]	0.004
Pre-hospital admission for COVID-19–Laboratory findings		
Hemoglobin (g/dL)	0.937 [0.842–1.043]	0.617
White blood cells (n/mm^3^)		.
<4500 (Ref.)	1	
4500–7000	0.751 [0.443–1.272]	0.287
>7000	0.797 [0.458–1.388]	0.423
Lymphocytes (n/mm^3^),		.
<700 (Ref.)	1	
700–1200	0.832 [0.507–1.367]	0.468
>1200	0.634 [0.357–1.125]	0.120
Platelets (n/mm^3^) (×100)	1.000 [0.998–1.003]	0.927
Neutrophils (n/mm^3^) (×100)	0.988 [0.912–1.071]	0.770
Albumin (g/dL)	0.762 [0.630–0.923]	0.005
C-reactive protein (mg/L)		
<5.6 (Ref.)	1	
5.6–24	2.492 [1.349–4.602]	0.004
>24	2.746 [1.495–5.043]	0.001
D-dimer (ng/mL)		.
<700 (Ref.)	1	
700–1500	1.02 [0.433–2.403]	0.963
>1500	2.175 [1.049–4.513]	0.037
Fibrinogen (mg/dL)		.
<400 (Ref.)		
400–600	1.37 [0.613–3.063]	0.444
>600	1.192 [0.51–2.786]	0.685
AST (IU/L)	0.999 [0.995–1.003]	0.678
Alanin aminotransferase (ALT) (IU/L)	0.998 [0.988–1.009]	0.681
GGT (IU/L)	0.997 [0.992–1.002]	0.870
LDH (IU/L)		.
<200 (Ref.)	1	
200–300	1.998 [ 0.922–4.331]	0.079
>300	1.798 [0.813–3.976]	0.147
Troponin T (ng/mL)	0.996 [0.982–1.01]	0.336
Procalcitonin (ng/mL)		.
<0.5 (Ref.)	1	
0.5–1	1.207 [0.479–3.04]	0.690
>1	2.48 [1.085–5.672]	0.031
IL-6 (ng/L)	0.998 [0.988–1.008]	0.744
Ferritin (μg/L)		.
<400 (Ref.)	1	
400–800	0.321 [1.092–0]	0.093
>800	1.218 [0.731–2.029]	0.449
Transferrin Saturation Index (%)	0.999 [0.984–1.015]	0.958
COVID-19–Outcomes		
Hospital admission	5.295 [1.937–14.470]	0.001
Oxygen saturation	0.857 [0.809–0.908]	<0.001
Non-invasive mechanical ventilation	2.177 [1.342–3.529]	0.002
ICU admission	1.560 [0.718–3.392]	0.262
Endotracheal intubation	1.727 [0.697–4.276]	0.238
COVID-19–Treatments		
Hydroxychloroquine	1.251 [0.692–2.263]	0.459
Azithromycin	1.268 [0.775–2.075]	0.344
Lopinavir/Ritonavir	1.938 [1.185–3.167]	0.008
Steroids	1.162 [0.738–1.831]	0.516
Beta-Interferon	3.663 [1.738–7.719]	0.001
Tocilizumab	0.872 [0.350–2.174]	0.769
Antibiotics	1.402 [0.841–2.336]	0.195
No Treatment	0.599 [0.218–1.647]	0.320

Expressed as percentages, mean ± SD or median [IQR p25–75]. SD: standard deviation; IQR: interquartile range; ACEI: angiotensin converting enzyme inhibitors; ARB; anti-AT1 receptor antagonists, NSAIDs: non-steroidal anti-inflammatory agents.

**Table 4 nutrients-13-02559-t004:** Clinical and analytical characteristics associated to mortality in the multivariate Cox regression model (death: 81/285).

	HR [IC95%]	*p*
Age (years)		
≤70 (Ref)	1	
70–80	2.911 [1.532–5.532]	0.001
≥80	2.372 [1.195–4.707]	0.014
Serum albumin (g/dL)	0.668 [0.496–0.900]	0.008
Paricalcitol use (Yes)	0.499 [0.278–0.898]	0.020
Calcimimetics use (Yes)	0.332 [0.12–0.921]	0.034
C-reactive protein (mg/dL)		
<5.6 (Ref)	1	
5.6–24	1.576 [0.808–3.072]	0.182
>24	2.417 [1.284–4.55]	0.006
PTH (ng/L)	0.999 [0.998–1.000]	0.167
Serum calcium (mg/dL)	1.046 [0.837–1.306]	0.695
Serum phosphorus (mg/dL)	1.037 [0.863–1.246]	0.697

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
