# Peer review of "Mortality in Hemodialysis Patients with COVID-19, the Effect of Paricalcitol or Calcimimetics"

_nutrients, 2021, doi:10.3390/nu13082559_

Round 1

Reviewer 1 Report

Dolores Arenas et al conducted an retrospective study to investigate the impact of VD status and the treatments of active vitamin D medications on outcomes of a cohort of COVID-19 positive HD patients. Overall, it is an important study which provided useful information for the management of HD patients with COVID-19 in clinical practice. However, these are some problems regarding how these patients were grouped. It is not appreciated to report outcomes based on what current groups did, such as the group of patients treated with calcitrio and those without, as the later group may receive paricalcitol or calcimimetics or both. Patients should be grouped as following: no active vitamin D medications, paricalcitol, calcimimetics, paricalcitol or calcimimetics, paricalcitol or calcimimetics. The baseline clinical characteristics and outcomes for each group should be presented.

Author Response

Revisor 1. Dolores Arenas et al conducted an retrospective study to investigate the impact of VD status and the treatments of active vitamin D medications on outcomes of a cohort of COVID-19 positive HD patients. Overall, it is an important study which provided useful information for the management of HD patients with COVID-19 in clinical practice. However, these are some problems regarding how these patients were grouped. It is not appreciated to report outcomes based on what current groups did, such as the group of patients treated with calcitrio and those without, as the later group may receive paricalcitol or calcimimetics or both. Patients should be grouped as following: no active vitamin D medications, paricalcitol, calcimimetics, paricalcitol or calcimimetics, paricalcitol or calcimimetics. The baseline clinical characteristics and outcomes for each group should be presented.

We added table 2. Baseline characteristics of patients treated with Active vitamin D medication. Mortality associated with COVID-19 was significantly lower in patients treated with any form of vitamin D (27/136, 19.9%) compared to untreated (54/149; 36.2%) p:0.002. Only patients treated with paricalcitol or calcimimetics showed significantly lower mortality compared to those who received calcidiol or calcitriol (table 3) (Supplementary table 2)

Then, we compared possible differences between baseline characteristics of COVID-19 HD patients treated with no active vitamin D medication, paricalcitol, calcimimetics, paricalcitol or calcimimetics and those who were not treated with either drug (Supplementary tables 2,3,4, 5 and 6, respectively).

Supplementary table 2. Baseline characteristics of patients treated with Active vitamin D medication and death associated to active Vit D medications. Total death: 81 patients. N: 285 .  

Active vitamin D

medication

N:136

Death: 27

Mortality : 19.9%

No active vitamin D medication

N= 149

Death= 54

Mortality=36.2%

p

Age (years, mean ± SD)

72.6 ± 13.0

72.0 ± 12.3

0.728

Female gender

45 (33.1%)

38 (25.5%)

0.101

Body mass index (kg/m2, mean ± SD)

26.4 ± 5.1

24.9 ± 5.7

0.021

Dialysis vintage (years, median [p25-75])

4.5 [3.5 – 5.5]

4.2 [3.3 -5.0]

0.626

Native fistulae as vascular access (N : 261)

70/125 (56%)

84/136 (61.8%)

0.206

Current Smoking status (N: 205)

    46/99 (46.5%)

54/106 (50.9%)

0.308

Alcohol (N: 177)

5/84 (6%)

12/93 (12.9%)

0.094

Previous transplant (N : 261)

12/125 (9.6%)

8 /136(5.9%)

0.185

Diabetes Mellitus

64/136(47.1%)

72/149 (48.3%)

0.462

Arterial hypertension

57/136(41.9%)

69/149 (46.3%)

0.265

Tuberculosis (N : 261)

2/125 (1.6%)

3/136 (2.2%)

0.540

Dyslipemia

45/136 (33.1%)

60/149(40.3%)

0.129

Lung disease

31/136 (22.8%)

35/149 (23.5%)

0.501

Ischemic cardiopathy

42/136 (30.9%)

51/149 (34.2%)

0.318

Calcidiol

32/136 (23.5%)

0/149 (0%)

0.000

Calcitriol

21/136 (15.4%)

0/149 (0%)

0.000

Paricalcitol

94/136 (69.1%)

0/149 (0%)

0.000

Calcimimetics

28/136 (20.6%)

21/149 (14.1%)

0.092

25-OH-D (ng/ml, , mean ± SD) ( N : 268)

20.9 ± 13.4

20.8 ± 13.5

0.972

PTH (ng/L, median [p25-75])

345 [291 - 398]

294 [238 - 349]

0.191

Calcium (mg/dl, mean ± SD)

8.8 ± 0.8

8.8 ± 0.6

0.834

Phosphate (mg/dl, mean ± SD)

4.2 ± 0.9

3.9 ± 1.2

0.054

ACEIs

32/136(23.5%)

29/149 (19.5%)

0.245

ARBs

12/136 (8.8%)

19/149 (12.8%)

0.192

NSAIDs

12/136 (8.8%)

9/149 (6.0%)

0.251

Albumin baseline (g/dl, mean ± SD) 

3.8 ± 0.3

3.7 ± 0.4

0.058

C-reactive protein (mg/L, median [p25-75]) at Hospital admission

60.7 [41-79]

45.8 [28-62]

0.249

Ageusia (N : 93)

1/38 (2.6%)

3/55(5.5%)

0.458

Anosmia( N: 93)

2/38(5.3%)

3/55(5.5%)

0.671

Cough (N : 265)

68/125(54.4%)

59/140 (42.1%)

0.031

Fever(N : 265)

76/125 (60.8%)

91/140 (65%)

0.281

Diarrhea(N : 265)

10/125 (8%)

18/140 (12.9%)

0.139

Dyspnea(N : 265)

39/125 (31.2%)

46/140 (32.9%)

0.438

Asymptomatic (N : 276)

17/132 (12.9%)

18/144 (12.5%)

0.534

Pneumonia ( N : 254)

87/122 (71.3%)

96/132 (72.7%)

0.455

Hospital admission

107/136 (78.7%)

121/149(81.2%)

0.350

ICU admission (N:257)

8/120 (6.7%)

7/137 (5.1%)

0.394

Endotracheal intubation (N : 254)

5/119 (4.2%)

5/135(3.7%)

0.545

Resolution and discharged(N : 264)

9/125 (72%)

79/139(56.8%)

0.007

Death

27/136 (19.9%)

54/149 (36.2%)

0.002

Hydroxychloroquine(N: 238)

82/112 (73.2%)

93/126 (73.8%)

0.517

Lopinavir/ Ritonavir (N: 239)

45/ 112 (40.2%)

47/127(37%)

0.354

Azithromycin( N: 239)

41/112(36.6%)

52/127(40.9%)

0.290

Steroids (N : 284)

41/136 (30.1%)

50/148 (33.8%)

0.299

Tocilizumab( N: 239)

9/112(8%)

11/127(8.7%)

0.525

Beta-Interferon( N: 239)

6/112 (5.4%)

6/127 (4.7%)

0.527

Remdesevir(N: 239)

0/112 (0.0%)

1/127 (0.8%)

0.531

Calcitriol

N:21

No calcitriol

N= 264

p

Death

4/21 (19%)

77/264 (29.2%)

0.235

Calcifediol

N:32

No calcifediol

N= 253

p

Death

10/32 (31.3%)

71/253 (28.1%)

0.424

Calcifediol or calcitriol

N:42

No calcifediol or calcitriol

N= 243

p

Death

11/42 (26.2%)

70/243 (28.8%)

0.854

Supplementary table 3. Baseline characteristics of patients treated with paricalcitol. Total Death: 81 patients; N: 285 

Paricalcitol

N=94

Death= 16

Mortality= 17%

No paricalcitol

N= 191

Death= 65

Mortality= 34%

p

Age (years, mean ± SD)

72.3 ± 13.1

72.4 ± 12.4

0.953

Female gender

31 (33%)

53 (27.3%)

0.322

Body mass index (kg/m2, mean ± SD)

26.8 ± 5.29

25.04 ± 5.47

0.012

Dialysis vintage (years, median [p25-75])

2.5 [3.6 – 6.3]

2.5 [3.3 - 4.7]

0.178

Native fistulae as vascular access (N: 261)

55/94 (58.5%)

99/167 (59.3%)

0.903

Current Smoking status (N: 208)

    31/64 (48.4%)

71/144 (49.3%)

0.908

Alcohol (N: 177)

1/61 (1.6%)

16/116 (13.8%)

0.009

Previous transplant (N: 261)

10/94 (10.6%)

10 /167(6.0%)

0.175

Diabetes Mellitus

42/94 (44.7%)

94/191 (49.2%)

0.471

Arterial hypertension

33/94(35.1%)

93/191 (48.7%)

0.030

Tuberculosis (N: 261)

1/94 (1.1%)

4/167 (2.4%)

0.451

Dyslipemia

28/94 (29.8%)

77/191(40.3%)

0.083

Lung disease

20/94 (21.3%)

46/191 (24.1%)

0.597

Ischemic cardiopathy

29/94 (30.9%)

64/191 (33.5%)

0.653

Calcidiol

4/94 (4.3%)

28/191 (14.7%)

0.009

Calcimimetics

26/94 (27.7%)

23/191 (12%)

0.001

25-OH-D (ng/ml, mean ± SD) (N: 252)

20.06 ± 13.7

21.1 ± 13.3

0.189

< 20 ng/ml

59/88 (67%)

90/164 (54.9%)

20-30 ng/ml

13/88 (14.8%)

39/164 (23.8%)

>30 ng/ml

16/88 (18.2%)

35/164(21.30%)

PTH (ng/L, median [p25-75])

386 [196 - 479]

285 [105 - 343]

0.013

Calcium (mg/dl, mean ± SD)

8.8 ± 0.8

8.8 ± 0.6

0.870

Phosphate (mg/dl, mean ± SD)

4.2 ± 0.9

3.9 ± 1.2

0.125

ACEIs

24/94 (25.5%)

37/191 (19.4%)

0.233

ARBs

7/94 (7.4%)

24/191 (12.6%)

0.192

NSAIDs

8/94 (8.5%)

13/191 (6.8%)

0.605

Albumin baseline (g/dl, mean ± SD) 

3.8 ± 0.4

3.7 ± 0.3

0.724

C-reactive protein (mg/L, median [p25-75]) at Hospital admission

77.2 [49.7-104.7]

41.5 [28.2-54.9]

0.010

Ageusia (N: 93)

1/15 (6.7%)

3/78(3.8%)

0.622

Anosmia (N: 93)

1/15 (6.7%)

4/78 (5.1%)

0.809

Cough (N: 265)

44/85 (51.8%)

83/180 (46.1%)

0.390

Fever (N: 265)

48/85 (56.5%)

119/180 (66.1%)

0.129

Diarrhea (N: 265)

4/85 (4.7%)

24/180 (13.3%)

0.033

Dyspnea (N: 265)

26/85 (30.6%)

59/180(32.8%)

0.722

Asymptomatic (N: 276)

13/91 (14.3%)

22/185 (11.9%)

0.574

Pneumonia (N: 254)

61/83 (73.5%)

122/171 (71.3%)

0.720

Hospital admission

74/94 (78.7%)

154/191(80.6%)

0.705

ICU admission (N:257)

6/81 (7.4%)

9/176 (5.1%)

0.466

Endotracheal intubation (N: 254)

5/81 (6.2%)

5/173(2.9%)

0.210

Resolution and discharged (N: 264)

64/83 (77.1%)

105/181(58%)

0.003

Death

16/94 (17.0%)

65/191 (34.0%)

0.003

Hydroxychloroquine (N: 238)

55/76 (72.4%)

120/162 (74.1%)

0.781

Lopinavir/ Ritonavir (N: 239)

25/ 76 (32.9%)

67/163(41.1%)

0.224

Azithromycin (N: 239)

21/76(27.6%)

72/163(44.2%)

0.015

Steroids (N: 284)

24/94 (25.5%)

67/190 (35.3%)

0.098

Tocilizumab (N: 239)

6/76(7.9%)

14/163(8.6%)

0.857

Beta-Interferon (N: 239)

4/76 (5.3%)

8/163 (4.9%)

0.907

Remdesevir (N: 239)

0/76 (0.0%)

1/163 (0.6%)

0.494

 *Resolution: discharged home at the time of reporting. Expressed as percentages, mean ± SD or median [IQR p25-75]. SD: standard deviation; IQR: interquartile range; ACEI: angiotensin converting enzyme inhibitors; ARB: AT1 receptor antagonists, NSAIDs: non-steroidal anti-inflammatory agents.

Supplementary table 4. Baseline characteristics of patients treated with calcimimetics. Total Death 81 patients. N: 285.

Calcimimetics

N= 49

Death= 4

Mortality = 8.2%

No calcimimetics

N= 236

Death= 77

Mortality = 32.6%

p

Age (years, mean ± SD)

67.9 ± 15.4

73.3 ± 11.8

0.023

Female gender

11/49 (22.4%)

72/236 (30.5%)

0.258

Body mass index (kg/m2, mean ± SD)

27.9 ± 6.8

25.1 ± 5.0

<0.001

Dialysis vintage (years, median [p25-75])

4.1 [2.3 -8.5]

2.3 [1.0-4.4]

<0.001

Native fistulae as vascular access (N: 261)

30/46 (65.2%)

124/215 (57.7%)

0.345

Current Smoking status (N: 205)

11 /31 (35.5%)

89/174 (51.1%)

0.108

Alcohol (N: 177)

3/28 (10.7%)

14/149(9.4%)

0.828

Previous transplant

6/46 (13%)

14/215 (6.5%)

0.131

Diabetes Mellitus

14 /49 (28.6%)

122/239 (51.7%)

0.003

Arterial hypertension

16/49 (32.7%)

110/236 (46.6%)

0.073

Tuberculosis

1/46 (2.2%)

4/215 (1.9%)

0.888

Dyslipemia

16/49 (32.7%)

89/236 (37.7%)

0.504

Lung disease

6/49 (12.2%)

60/236(25.4%)

0.047

Ischemic cardiopathy

18/49 (36.7%)

75/236(31.8%)

0.501

Calcidiol

0/49 (0%)

32/236 (13.6%)

0.006

Calcitriol

2/49 (4.1%)

19/236 (8.1%)

0.333

Paricalcitol

26/49 (53.1%)

68/236 (28.8%)

0.001

Calcitriol and/or Paricalcitol

28/49 (57.1%)

83/236(35.2%)

0.004

25-OH-D (ng/ml, mean ± SD) (N: 252)

19.6 ± 13.2

21.0 ± 13.5

0.514

PTH (ng/L, median [p25-75])

434 [ 301-685]

210 [113-333]

< 0.001

Calcium (mg/dl, mean ± SD)

8.6 ± 0.7

8.8 ± 0.7

0.070

Phosphate (mg/dl, mean ± SD)

4.3 ± 1.1

3.9 ± 1.1

0.044

Angiotensin converting enzyme inhibitors

12/49 (24.5%)

49/236(20.8%)

0.563

Anti-AT1 receptor antagonists

3/49 (6.1%)

28/236 (11.9%)

0.240

Non-steroidal anti-inflammatory agents

7/49 (14.3%)

14/236 (5.9%)

0.042

Albumin baseline (g/dl, mean ± SD)

3.92 ± 0.37

3.78 ± 0.39

0.041

Hospital admission C-reactive protein (mg/L, median [p25-75])

34 [18-51]

43 [32-53]

0.792

Ageusia (N: 93)

0/11 (0%)

4/82 (4.9%)

0.454

Anosmia (N: 93)

0/11 (0%)

5/82 (6.1%)

0.400

Cough (N: 265)

21/46 (45.7%)

106/219(48.4%)

0.734

Fever (N: 265)

24/46 (65.3%)

143/219 (52.2%)

0.094

Diarrhea (N: 265)

2/46 (4.3%)

26/219 (11.9%)

0.131

Dyspnea (N: 265)

11/46 (23.9%)

74/219 (33.8%)

0.192

Asymptomatic (N: 276)

7/47 (14.9%)

28/229 (12.2%)

0.617

Pneumonia (N: 254)

31/45 (68.9%)

152/209 (72.7%)

0.603

Hospital admission (n, %)

38/49 (77.6%)

190 / 236 (80.5%)

0.638

ICU admission (n, %) (N: 257)

3/43 (7%)

12/214 (5.6%)

0.727

Non-invasive mechanical ventilation (n, %)

(N: 220)

11/32 (34.4%)

63/188 (33.5%)

0.924

Endotracheal intubation (n, %)

2/43 (4.7%)

8/211 (3.8%)

0.792

Resolution and discharged

42/46 (91.3%)

127/218 (58.3%)

<0.001

Death

4/49 (8.2%)

77/236 (32.6%)

0.001

Hydroxychloroquine (N: 238)

32/42 (76.2%)

143/196 (73%)

0.667

Lopinavir/ Ritonavir (N: 239)

12/42 (28.6%)

80/197 (40.6%)

0.145

Steroids (N: 284)

10/48 (20.8%)

81/236(34.3%)

0.068

Azithromycin (N: 239)

17/42 (40.5%)

76/197 (38.6%)

0.819

Tocilizumab (N: 239)

3/42 (7.1%)

17/197(8.6%)

0.752

Beta-Interferon (N: 239)

1/42 (2.4%)

11/197 (5.6%)

0.388

Remdesevir (N: 239)

0/42 (0%)

1/197 (0.5%)

0.644

*Resolution: discharged home at the time of reporting. Expressed as percentages, mean ± SD or median [IQR p25-75]; SD: standard deviation; IQR: interquartile range; ACEIs: angiotensin converting enzyme inhibitors; ARBs: AT1 receptor antagonists, NSAIDs: non-steroidal anti-inflammatory agents.

Supplementary table 5. Baseline characteristics of patients treated with calcimimetics or paricalcitol. Total Death= 81 patients; N: 285

Calcimimetics or paricalcitol

N=117

Death= 19

Mortality = 16.2%

No calcimimetics or paricalcitol

N=168

Death= 62

Mortality = 36.9%

p

Age (years, mean ± SD)

71.4 ± 13

72.9 ± 12.3

0.342

Female gender

33/117 (28.2%)

50/168 (29.8%)

0.776

Body mass index (kg/m2, mean ± SD)

27.1 ± 6.1

24.6 ± 4.6

<0.001

Dialysis vintage (years, median [p25-75])

5.2 [4.0-6.3]

3.8 [3.0-4.5]

0.039

Native fistulae as vascular access

67/114 (58.8%)

87/147 (59.2%)

0.947

Current Smoking status (N: 205)

38/79 (48%)

62/126 (49.2%)

0.878

Alcohol (N: 177)

4/74 (5.4%)

13/103(12.6%)

0.108

Previous transplant (N: 261)

12/114 (10.5%)

8/147 (5.4%)

0.126

Diabetes Mellitus

47/117 (40.2%)

89/117 (53%)

0.033

Arterial hypertension

40/117 (34.2%)

86/168 (51.2%)

0.004

Tuberculosis (N: 261)

1/114 (0.9%)

4/147(2.3%)

0.281

Dyslipemia

37/117 (31.6%)

68/168 (40.5%)

0.128

Lung disease

24/117 (20.5%)

42/168(25%)

0.377

Ischemic cardiopathy

38/117 (32.5%)

55/168 (32.7%)

0.963

Calcidiol

4/117 (3.4%)

28/168 (16.7%)

<0.001

Calcitriol

6/117 (5.1%)

15/168 (8.9%)

0.227

25-OH-D (ng/ml, median [p25-75])

19.7 [17.1-22.2]

21.8 [19.3-23.8]

0.219

PTH (ng/L, median [p25-75])

436 [ 363-508]

231.3 [197-266]

< 0.001

Calcium (mg/dl, mean ± SD)

8.7 ± 0.8

8.8 ± 0.6

0.497

Phosphate (mg/dl, mean ± SD)

4.24± 1.18

3.9 ± 1.04

0.033

Angiotensin converting enzyme inhibitors

27/117 (23.1%)

34/168(20.2%)

0.565

Anti-AT1 receptor antagonists

8/117 (6.8%)

23/168 (13.7%)

0.068

Non-steroidal anti-inflammatory agents

12/117 (10.3%)

9/168 (5.4%)

0.119

Hospital admission Albumin (g/dl, mean ± SD)

3.8 ± 0.5

3.7± 0.51

0.043

Hospital admission C-reactive protein (mg/L, median [p25-75])

66.5 [44.8-88]

43.5 [27.9-59.1]

0.080

Ageusia (N: 93)

1/22 (4.5%)

3/71 (4.1%)

0.948

Anosmia (N: 93)

1/22 (4.5%)

4/71 (5.5%)

0.843

Cough (N: 265)

55/108 (50.9%)

72/157(45.9%)

0.417

Fever (N: 265)

60/108 (55.6%)

107/157 (68.2%)

0.027

Diarrhea (N: 265)

6/108 (5.6%)

22/157 (14%)

0.028

Dyspnea (N: 265)

34/108 (31.5%)

51/157 (32.5%)

0.864

Asymptomatic (N: 276)

16/114 (14%)

19/162 (11.7%)

0.571

Pneumonia (N: 254)

77/106 (72.6%)

106/148(71.6%)

0.858

Hospital admission (n, %)

93/117 (79.5%)

135/168(80.4%)

0.857

ICU admission (n, %) (N: 257)

8/103 (7.8%)

7/154 (4.5%)

0.282

Non-invasive mechanical ventilation (n, %) (N: 220)

31/84 (36.9%)

43/136 (31.6%)

0.420

Endotracheal intubation (n, %) (N: 254)

6/103 (5.8%)

4/151 (2.6%)

0.201

Resolution and discharged (N: 264)

83/105 (79%)

86/159 (54.1%)

<0.001

Death

19/117 (16.2%)

62/168 (36.9%)

<0.001

Hydroxychloroquine (N: 238)

72/97 (74.2%)

103/141 (73%)

0.840

Lopinavir/ Ritonavir (N: 239)

31/97 (32%)

61/142 (43 %)

0.086

Steroids (N: 284)

29/116 (25%)

62/168 (36.9%)

0.035

Azithromycin (N: 239)

31/97 (32%)

62/142 (43.7%)

0.068

Tocilizumab (N: 239)

8/97 (8.2%)

12/142(8.5%)

0.956

Beta-Interferon (N: 239)

4/97 (4.1%)

8/142 (5.6%)

0.600

Remdesevir (N: 239)

0/97 (0%)

1/142 (0.7%)

0.408

*Resolution: discharged home at the time of reporting. Expressed as percentages, mean ± SD or median [IQR p25-75].SD: standard deviation; IQR: interquartile range; ACEIs: angiotensin converting enzyme inhibitors; ARBs: AT1 receptor antagonists, NSAIDs: non-steroidal anti-inflammatory agents.

Supplementary table 6. Baseline characteristics of patients treated with paricalcitol and calcimimetics. Total Death=81 patients; N: 285;  X2 =8.4, p:0.004.

Calcimimetics and paricalcitol

N=26

Death= 1

Mortality = 3.8%

others

N=259

Death= 80

Mortality = 30.9%

p

Age (years, mean ± SD)

67.6 ± 17

72.8 ± 11.9

0.046

Female gender

9/26 (34.6%)

74/259 (28.6%)

0.518

Body mass index (kg/m2, mean ± SD)

27.4 ± 4.6

25.4 ± 5.5

0.079

Dialysis vintage (years, median [p25-75])

6.3 [3.6-9]

4.1 [3.5-4.8]

0.056

Native fistulae as vascular access (N: 261)

18/26 (69.2%)

136/235 (57.9%)

0.264

Current Smoking status (N: 205)

4/16 (25%)

96/189(50.8%)

0.047

Alcohol (N: 177)

0/15 (0%)

17/162(10.5%)

0.187

Previous transplant (N: 261)

4/26 (15.4%)

16/235 (6.8%)

0.119

Diabetes Mellitus

9/26 (34.6%)

127/259 (49%)

0.161

Arterial hypertension

9/26 (34.6%)

117/259 (45.2%)

0.301

Tuberculosis

1/26 (1.7%)

4/235(3.8%)

0.449

Dyslipemia

7/26 (26.9%)

98/259 (37.8%)

0.271

Lung disease

2/26 (7.7%)

64/259(24.7%)

0.050

Ischemic cardiopathy

9/26 (34.6%)

84/259 (32.4%)

0.821

Calcidiol

0/26 (0.0%)

32/259 (12.4%)

0.057

Calcitriol

0/26 (0.0%)

21/259 (8.1%)

0.131

25-OH-D (ng/ml, median [p25-75])

20.7 [14.8-26.7]

20.9 [19.1-22.6]

0.963

PTH (ng/L, median [p25-75])

514 [ 381-647]

298 [258-337]

< 0.001

Calcium (mg/dl, mean ± SD)

8.7 ± 0.8

8.8 ± 0.6

0.335

Phosphate (mg/dl, mean ± SD)

4.4± 1.18

4.0 ± 1.04

0.118

Angiotensin converting enzyme inhibitors

9/26 (34.6%)

52/259(20.1%)

0.085

Anti-AT1 receptor antagonists

2/26 (7.7%)

29/259 (11.2%)

0.584

Non-steroidal anti-inflammatory agents

3/26 (11.5%)

18/259 (6.9%)

0.393

Hospital admission Albumin (g/dl, mean ± SD)

3.33 ± 0.5

3.31± 0.51

0.957

Hospital admission C-reactive protein (mg/L, median [p25-75])

54.2 [11.4-97.0]

52.9 [39.4-66.4]

0.070

Ageusia (N: 93)

0/4 (0%)

4/89 (4.5%)

0.665

Anosmia (N: 93)

0/4 (0%)

5/89 (5.6%)

0.626

Cough (N: 265)

10/23 (43.5%)

117/242(48.3%)

0.655

Fever (N: 265)

12/23 (52.2%)

155/242 (64%)

0.260

Diarrhea (N: 265)

0/23 (0%)

28/242 (11.6%)

0.085

Dyspnea (N: 265)

3/23 (13%)

82/242 (33.9%)

0.041

Asymptomatic (N: 276)

4/24 (16.7%)

31/252 (12.3%)

0.539

Pneumonia (N: 254)

15/22 (68.2%)

168/232(72.4%)

0.672

Hospital admission (n, %)

19/26 (73.1%)

209/259(80.7%)

0.355

ICU admission (n, %) (N: 257)

1/21 (7.8%)

14/236 (5.1%)

0.826

Non-invasive mechanical ventilation (n, %)

(N: 220)

5/15 (33.3%)

69/205 (33.7%)

0.979

Endotracheal intubation (n, %) (N: 254)

1/21 (4.8%)

9/233 (3.9%)

0.839

Resolution and discharged (N: 264)

23/24 (95.8%)

146/240 (60.8%)

<0.001

Death

1/26 (3.8%)

80/259 (30.9%)

<0.001

Hydroxychloroquine (N: 238)

15/21 (71.4%)

160/217 (73.7%)

0.819

Lopinavir/ Ritonavir (N: 239)

6/21 (28.6%)

86/218 (39.4 %)

0.328

Steroids (N: 284)

5/26 (19.2%)

86/258 (33.3%)

0.142

Azithromycin (N: 239)

7/21 (33.3%)

86/218 (39.4%)

0.583

Tocilizumab (N: 239)

1/21 (4.8%)

19/218(8.7%)

0.532

Beta-Interferon (N: 239)

1/21 (4.8%)

11/218 (5.0%)

0.955

Remdesevir (N: 239)

0/21 (0%)

1/218 (0.5%)

0.756

*Resolution: discharged home at the time of reporting. Expressed as percentages, mean ± SD or median [IQR p25-75].SD: standard deviation; IQR: interquartile range; ACEIs: angiotensin converting enzyme inhibitors; ARBs: AT1 receptor antagonists, NSAIDs: non-steroidal anti-inflammatory agents.

Reviewer 2 Report

It was with interest that I read María Dolores Arenas et al. manuscript “Mortality in hemodialysis patients with COVID-19, the effect of paricalcitol or calcimimetics”.

The authors evaluated the impact of vitamin status and the administration of vitamin D calcitriol and calcimimetics, used to treat secondary hyperparathyroidism, on survival of a cohort of 288 hemodialysis spanish patients affected by COVID-19, followed for 14-35 days or until death.

The authors found that the use of paricalcitol, calcimimetics or the combination of both, seems be associated to improved survival in HD patients with COVID-19.

This interesting manuscript investigates a new and full of therapeutic expectations on the optimal therapy in HD patients, frail patients affected by a high mortality rate.

The prevalence of COVID-19 in HD (6,1%) was not very high, according to other experience, but less than reported by other authors (more than 15%), 80% of whome were hospitalized because symptomatic (fever in 68% and dyspnea 32%). All the patients underwent to the optimal empiric therapy at the time of the first wave of pandemic (hydroxychloroquine, azithromycin, lopinavir/ritonavir, steroids, and even tocilizumab (in 8,7%) before subsequent studies questioned all the empirical therapies used in the first phase of the pandemic. No significant differences between treated and not treated patients resulted in hospitla admission, non invasive mechanical ventilation or intubation. Of note that increased survival were observed in HD patients treated with paricalcitol and/or calcimimetics and even in those with higher serum albumin levels, probably because younger and better nutrished.

The improved survival was attribuited to plausible pleiotropic effects related to anti-inflammatory, anti-viral and anti-hypotensive properties: Vitamin D and drugs agent on mineral bone disease also have a variety of effects unrelated with mineral and bone metabolism, including the regulation of arterial blood pressure and the prevention of cardiovascular complications, modulation of immunological responses, regulation of insulin production and prevention against diabetes, protection against certain cancers and other beneficial actions.

However, the authors have to better deepen this aspect, keystone of the manuscript, also because the literature is poor on this topic.

Minor observations:

  • page 5 “KDIGO and Spanish guidelines in CKD MBD, please add references
  • Table 1, a relatively high prevalence (41%) of patients had a CVC as vascular access, related to worst outcome compared with AVF
  • Table 1, comorbidities: 47,6% of HD patients had diabetes, is this data similar to spanish picture?
  • Table 1: Phosphate (4,1 mg/dL), calcium (8,8 mg/dL) and PTH (240 pg/dL) levels within the limits reflects an optimal control of MBD

Author Response

It was with interest that I read María Dolores Arenas et al. manuscript “Mortality in hemodialysis patients with COVID-19, the effect of paricalcitol or calcimimetics”.

The authors evaluated the impact of vitamin status and the administration of vitamin D calcitriol and calcimimetics, used to treat secondary hyperparathyroidism, on survival of a cohort of 288 hemodialysis spanish patients affected by COVID-19, followed for 14-35 days or until death.

The authors found that the use of paricalcitol, calcimimetics or the combination of both, seems be associated to improved survival in HD patients with COVID-19.

This interesting manuscript investigates a new and full of therapeutic expectations on the optimal therapy in HD patients, frail patients affected by a high mortality rate.

The prevalence of COVID-19 in HD (6,1%) was not very high, according to other experience, but less than reported by other authors (more than 15%), 80% of whome were hospitalized because symptomatic (fever in 68% and dyspnea 32%). All the patients underwent to the optimal empiric therapy at the time of the first wave of pandemic (hydroxychloroquine, azithromycin, lopinavir/ritonavir, steroids, and even tocilizumab (in 8,7%) before subsequent studies questioned all the empirical therapies used in the first phase of the pandemic. No significant differences between treated and not treated patients resulted in hospitla admission, non invasive mechanical ventilation or intubation. Of note that increased survival were observed in HD patients treated with paricalcitol and/or calcimimetics and even in those with higher serum albumin levels, probably because younger and better nutrished.

The improved survival was attribuited to plausible pleiotropic effects related to anti-inflammatory, anti-viral and anti-hypotensive properties: Vitamin D and drugs agent on mineral bone disease also have a variety of effects unrelated with mineral and bone metabolism, including the regulation of arterial blood pressure and the prevention of cardiovascular complications, modulation of immunological responses, regulation of insulin production and prevention against diabetes, protection against certain cancers and other beneficial actions.

However, the authors have to better deepen this aspect, keystone of the manuscript, also because the literature is poor on this topic.

As requested, we have introduced in the discussion, the most recent advances in our understanding of the mechanisms underlying vitamin D actions against Covid19 infection and on the doses required to ameliorate disease progression and outcomes.

Minor observations:

page 5 “KDIGO and Spanish guidelines in CKD MBD, please add references

We introduce bibliographic references 17 and 18

  1. Kidney Disease: Improving Global Outcomes (KDIGO) CKD-MBD Update Work Group. KDIGO 2017 Clinical Practice Guideline Update for the Diagnosis, Evaluation, Prevention, and Treatment of Chronic Kidney Disease-Mineral and Bone Disorder (CKD-MBD) [published correction appears in Kidney Int Suppl (2011). 2017 Dec;7(3):e1]. Kidney Int Suppl (2011). 2017;7(1):1-59.
  2. José-Vicente Torregrosa , Jordi Bover , Jorge Cannata Andía , Víctor Lorenzo , ALM de Francisco , Isabel Martínez , Mariano Rodríguez Portillo , Lola Arenas , Emilio González Parra, Francisco Caravaca, Alejandro Martín-Malo , Elvira Fernández Giráldez, Armando Torres. Recomendaciones de la Sociedad Española de Nefrología para el manejo de las alteraciones del metabolismo óseo-mineral en los pacientes con enfermedad renal crónica (S.E.N.-MM). Nefrologia 2011;31(Suppl.1):3-32
  • Table 1, a relatively high prevalence (41%) of patients had a CVC as vascular access, related to worst outcome compared with AVF

Table 3 shows the results on vascular access survival. The presence of a catheter does not influence survival from COVID in this population

Table 3. Clinical characteristics and associations with mortality in univariate Cox regression analyses

Native fistulae as vascular access

0.679 [0.428-1.078]

0.101

  • Table 1, comorbidities: 47,6% of HD patients had diabetes, is this data similar to spanish picture?

The average number of patients in HD with DM is around a 16-17% in the Spanish Registry. In the Spanish Covid 19 registry, 22% of Covid 19 patients had DM.The prevalence of DM in this cohort is higher than that reported for the whole country. It is important to emphasize that there are major differences in DM according to the geografic área of the country, and this particular HD cohort includes the Canary Islands,one of the área with the highest incidence of DM in Spain.

Table 1: Phosphate (4,1 mg/dL), calcium (8,8 mg/dL) and PTH (240 pg/dL) levels within the limits reflects an optimal control of MBD

We include in discussion :“Demographic parameters, comorbidities and baseline treatments and analytics in COVID-19 HD patients are shown in Table 1. Phosphate (4,1 mg/dL), calcium (8,8 mg/dL) and PTH (240 pg/dL) levels within the limits reflects an optimal control of MBD (17,18(”

Reviewer 3 Report

The authors are presenting very soundful explanations and concepts of the posssible protective effects of paricalcitol or calcimimetics regarding COVID mortality in a hemodialysis population.

There is novelty in this paper, but a weak point remains the limited patient number.

Spacing in the paper should be improved.

The abbreviation MDH is missing. It seems that "MHD" should be replaced with "HD".

In the results section (page 9 lower part): "Variables associated to...of secondary hyperpatathyroidisms" seems to be a heading (not evidenced).

As mentioned in page 10 (results section lower part). Patients treated with calcimetics were younger but presented a longer dialyis vintage. This fact needs to be discussed.

Figure 1 should be improved as the visual difference between the survival lines is too low (change greytones).

Discussion section:

Page 15-16: ...On the other hand, it is well known...and miR-30c [30, 49] might be cancelled.

The importance of dialysis vintage, patient age and its asssociation to calcimetic therapy should be discussed.

Author Response

Reviewer 3.

The authors are presenting very soundful explanations and concepts of the possible protective effects of paricalcitol or calcimimetics regarding COVID mortality in a hemodialysis population.

We check English language

There is novelty in this paper, but a weak point remains the limited patient number.

We include: “ The cross-sectional and retrospective nature of this analysis has limitations such as the the limited patient number”

Spacing in the paper should be improved.

Done

The abbreviation MDH is missing. It seems that "MHD" should be replaced with "HD".

Done

In the results section (page 9 lower part): "Variables associated to...of secondary hyperpatathyroidism" seems to be a heading (not evidenced).

This is the tittle of this section . We delete it

As mentioned in page 10 (results section lower part). Patients treated with calcimetics were younger but presented a longer dialyis vintage. This fact needs to be discussed.

See below

Figure 1 should be improved as the visual difference between the survival lines is too low (change greytones).

We change greytones in figures
